# Towards an Evidence-Based Model of Workplace Postvention

**DOI:** 10.3390/ijerph20010142

**Published:** 2022-12-22

**Authors:** Alison Clements, Angela Nicholas, Karen E Martin, Susan Young

**Affiliations:** 1School of Allied Health, University of Western Australia, Crawley, WA 6009, Australia; 2Melbourne School of Population and Global Health, University of Melbourne, Melbourne, VIC 3010, Australia; 3School of Education, University of Tasmania, Hobart, TAS 7005, Australia; 4Social Work and Social Policy, University of Western Australia, Crawley, WA 6009, Australia

**Keywords:** suicide, postvention, bereavement, funerals, workplaces, health and safety

## Abstract

This instrumental case study explored what suicide postvention might offer workplaces using the example of a large metropolitan funeral company. A mixed methods approach was utilized to examine staff experiences with suicide bereavement funerals and responses to a bespoke postvention training package. Staff found funerals due to suicide difficult in terms of communication, engagement and emotionality. These challenges were commonly characterized by increased tension and concern. In the absence of a postvention informed approach, staff had developed individual ways to negotiate the identified challenges of this work. The introduction of a staff-informed postvention training package delivered improvements in staff confidence with communication, understanding and management of the impact of suicide bereavement, and increased willingness to share information about postvention services with families and mourners. The findings indicated that benefits of the training could be extended through organizational governance and integration of supports. The findings are used to inform a model of workplace postvention together with a methodology incorporating staff experience and organizational context.

## 1. Introduction

“Bereavement by suicide is a unique form of grief characterized by features such as stigma, shame, and rejection that may complicate the grieving process and place people at heightened risk for specific mental health disorders, suicide attempts, and dying by suicide.” [1] (p. 1)

Suicide postvention is the formal and informal help and support provided to people bereaved by suicide [2]. Suicide postvention is a vital, though much neglected, aspect of suicide prevention, as evidence shows suicide bereavement is associated with impaired social functioning, poor mental and physical health outcomes and increased risk of suicidal ideation and behaviour [3,4]. It has been estimated that over half of the Australian population knows someone who has died by suicide [5] and the case has been made [6,7] for a public health approach to postvention. In this context, workplaces can play a dual role in mitigating risks associated with suicide bereavement, by:informing staff about working effectively with clients/customers/colleagues bereaved by suicide; andsupporting employees with the impact of work that exposes them to suicide and suicide-related bereavement.

Most research in this area has been conducted with emergency and healthcare workers [8]. It has been established that staff doing work involving exposure to sudden or violent death may develop a range of negative psychological and emotional symptoms variously characterised as burnout, compassion fatigue, and vicarious trauma or secondary traumatic stress [9]. Kimbrel et al. [10] found that increased risk of suicidality among fire fighters may relate to workplace exposure to suicide. 

Research shows that the negative impacts of workplace exposure to suicide are perpetuated by poor organisational preparedness. For example, an Australian study on mental health workers showed that “systemic issues [lack of internal processes, supports and resources] contribute to distress among exposed workers” [11] (p. 286). In a qualitative study on the experiences of English Ambulance staff attending incidents of suicide, participants reported a lack of training, support and workplace acknowledgement of the impact of suicide on workers [12]. Other studies have identified the distress and powerlessness experienced by staff responding to suicide [13,14] is due in part to their lack of training and resources [15,16].

Workplace culture and role expectations also play an important part in the impact of suicide on workers. For example, frontline healthcare staff report a reluctance to discuss their stress and distress because this conflicts with the ‘professional’ expectations of the role and workplace [17]. Ambulance staff report “supressing their distress despite significant emotional impact” of exposure to suicide [12]. It would appear that where there are no clear workplace postvention processes in place to mediate the impact of this work, staff are likely to have inhibitions about raising their needs or experiences.

In the UK, significant work has progressed to better support health care staff, General Practitioners, and frontline responders in responding to suicide bereavement [18]. In Australian public policy, there has been a somewhat ad hoc approach to postvention and there are few workplace-specific suicide postvention frameworks in operation. The primary example is ‘Be You’, delivered by headspace and Beyond Blue, which offers a comprehensive school-based postvention framework and includes relevant information, communications templates and tools for support [19]. Schools can self-select whether or not to engage with the framework and the extent to which its different components are implemented. Non-school based workplaces could perhaps apply modified versions of the ‘Be You’ framework but are not currently incentivized, resourced or supported to do so. The application to workplaces of evidence based postvention tools or approaches is not prioritized in current Australian suicide prevention and postvention frameworks (e.g., Postvention Australia Guidelines (2017) [2] Suicide Prevention: A Competency Framework (2021) [20] NSW Strategic Framework for Suicide Prevention 2018–202 (New South Wales 2018) [21] Victorian Suicide Prevention Framework 2016–2025 (Victoria 2016) [22]).

Viewed with a public health lens, and beyond first responder and health care roles, many workplaces will have staff exposed to suicide, particularly those dealing with high-risk populations such as men, Aboriginal and Torres Strait Islander populations, youth, older adults, culturally and linguistically diverse populations, indigenous populations, rural and remote populations, people with mental illness and or substance-related disorders, people with physical illnesses, people bereaved by suicide, prisoners and people experiencing socio-economic distress [23,24].As community stakeholders, workplaces can play a key role in a public health approach to postvention by equipping and supporting their staff to work safely and well in this space. In terms of their legislated Workplace Health and Safety requirements to mitigate psychological hazards for staff [25], workplaces also have a responsibility to establish clear processes and supports to mitigate the impact of suicide on staff.

Considered in this context, staff are a natural resource. Their lived experience (“personal knowledge about the world gained through direct, first-hand involvement in everyday events”) [26] is multidimensional and covers personal, workplace and consumer/client knowledge. 

In order to explore this approach, we conducted an Instrumental Case Study with staff from a major metropolitan funeral company in Perth, Western Australia (the name of the Company will be kept anonymous to protect participant identity). In Australia, funeral staff are the most commonly utilised service following a death [27]. There is currently no evidence for what funeral staff do, or do well, in the context of suicide bereavement. To date, there has been little acknowledgement or consideration of the potentially important role funeral service staff may play in postvention, or of the impact of this work on staff.

The research sought to answer the questions: ○What was the experience of funeral staff with funerals due to suicide? ○What was their response to a bespoke postvention training program [28] (see ‘Training’ designed with and for staff to assist them in this work?

The researcher in this Case Study took the role of both participant and observer. That is, she is employed by an agency (Neami National, a community based mental health organisation) in a role involving participation in an interagency postvention network. This provided her with access to a representative of the funeral company. Their discussions prompted the researcher to suggest the collaboration which led to the development of a staff training package (by Neami National) and initiation of this research study. The researcher did not write or deliver the training but was present as an observer during training sessions.

This is the first time such an exploration has occurred, and that a postvention approach has been applied and examined in this context. Findings have potential for application to the broader funeral industry -and for the development of a model of workplace postvention to support other workplaces and staff exposed to suicide.

## 2. Materials and Methods

The research was conducted in the form of an Instrumental Case Study, a methodology which “uses a particular case to gain appreciation of a broader issue or phenomenon” [29] (p. 2) and allows “interventions…and programme-based service reforms to be studied in detail in a real-life context” [30]. (p. 8) the research was approved by the University of Western Australia’s Human Research Ethics Committee Approval No: RA/4/20/5864.

The study sample was selected from a single family-owned funeral company with approximately 100 staff located at several sites across a metropolitan area. The name of the organisation is not reported here to maintain participant anonymity. 

A mixed methods approach was employed, with anonymous baseline and post-training questionnaires conducted via Qualtrics and semi-structured interviews conducted with between January 2020 and February 2021. 

### 2.1. Recruitment

In January 2020, all company staff (*n* = 100), were invited through direct email from researchers to complete the baseline questionnaire. The management selected 40 personnel to participate in four face to face postvention training sessions. Staff who participated in the training (‘Trainees’) were selected to represent a range of roles and office locations. Following each training session, the researchers emailed invitations to each Trainee to participate in an individual, face to face semi-structured interview. Post training questionnaires were distributed by the researchers to individual Trainees within two weeks of their training. 

### 2.2. Training

The training provided information on the extent and impacts of suicide bereavement, recommended language, stigma, boundaries, and resources and supports available to staff and clients following bereavement by suicide. The aim was to improve staff awareness, skills and confidence when working with clients bereaved by suicide and managing the impact of this work.

Training sessions ran for three hours and attended by mixed groups of between eight and 12 staff from different roles and office locations. The Training was delivered by Neami National staff and a professional trainer. The approach was a combination of didactic and discursive with trainees encouraged to share their experiences and responses together with each other and the group. 

### 2.3. Online Questionnaires

All company staff were invited to complete an eight-item baseline questionnaire prior to the Training, and Trainees were asked to complete a post-training questionnaire. 

The baseline questionnaire asked about experiences including the comparative difficulty and emotional impact of working with suicide bereavement funerals. The post questionnaire asked about the impact of the Training including confidence, awareness and willingness to share postvention information. Questionnaires were distributed and reported in Qualtrics software and data were recorded in Likert scales. To protect participant anonymity given the small sample size and context, baseline and post-Training surveys were not matched.

### 2.4. Semi-Structured Interviews

The semi-structured interview protocol included questions on the same variables measured in the questionnaires. It also included questions on period of employment with the company, role description and asked for more detail on staff experience working with suicide bereavement funerals. It was designed to elicit greater detail on the qualities of this experience and responses to the Training. 

Interview questions were deliberately open-ended and, as a result, discussions often led into unexpected and interesting areas. The researcher did not attempt to alter this natural flow; sometimes because the new material was relevant and revealing and sometimes because it was clearly important to the interviewee, and disrupting would have meant weakening rapport and flow.

The interview data were analysed utilising the iterative and inductive approach to thematic data coding outlined by Bazeley [31]. This requires close and repeated reading, reflection on and revision-of the interview data whilst developing individual ‘codes’ to represent the discrete units of meaning conveyed in the qualitative material. This process continues until codes can be further grouped together to adequately reflect categories of qualitative material, or ‘themes’, which are consistent and comparable across the data set. Analysis resulted in identifying three themes which are detailed in ‘Results’.

## 3. Results

### 3.1. Demographics

Twenty-three women and 13 men working in different roles and office locations participated in the training. The trainees had come to the company from a range of previous employment backgrounds and had been with the company for an average of seven years. The most recently employed trainee had been with the company for six months and the longest serving had been employed for over 30 years. Of the trainees, six women and five men participated in individual interviews. Any further identifying data was not collected to protect the anonymity of participants.

### 3.2. Baseline and Post-Training Questionnaires 

The baseline questionnaire was completed by two thirds (*n* = 67) of all company staff. Their responses, summarised in Table 1, showed that, compared with other types of funerals, three quarters of staff (74.6%) found those involving suicide to be more difficult and more complex in terms of communication. More than half (56.7%) of the staff found these funerals to have a greater emotional impact on themselves, and nearly two thirds (62.7%) said they had a greater emotional impact on colleagues. The results also indicated that although more than three quarters (82%) of staff were aware of suicide bereavement resources, only one quarter (26%) of staff had shared information or resources pertaining to suicide bereavement with families or mourners.

### 3.3. Post-Training Questionnaire Results

Of the 36 Trainees, 32 (88.9%) completed the post-Training questionnaire. The post-Training questionnaire data (Table 2) showed an overall positive effect of the Training. All staff (100%) surveyed found the Training useful and nearly all (85%) said they were more confident that they knew and used the appropriate (best practice) language when working with suicide bereavement funerals. All staff (100%) said the Training helped them better understand the emotional impact of working with suicide bereavement funerals and almost all staff (88%) said it helped them better manage this impact. The poorest outcome recorded pertained to the amount of pressure staff feel working with suicide bereavement funerals, with nearly half (43.8%) reporting no change as a result of the training. Almost all staff (85%) who completed the questionnaire had a better awareness of suicide bereavement services and were more comfortable (94%) with the idea of sharing this information with families and mourners. This finding is important given that the initial questionnaire showed only one quarter (26%) of staff had shared such information in the course of their work.

### 3.4. Semi-Structured Interviews—Thematic Analysis

Eleven staff participated in the semi-structured interviews. Interview data reiterated that the experience of staff working with suicide bereavement funerals was more complex, emotionally demanding and difficult in terms of communication and engagement with families and mourners, than other funerals. Individual interviewees reflected on these complexities in more detail and provided examples to illustrate the multiple dimensions of practice, adjustments and impacts of working with suicide bereavement funerals. 

There was also a commonly expressed anxiety amongst staff characterised by many as ‘walking on eggshells’. This was associated with the grief responses of families and the open- or closed-communication styles of the bereaved around the cause of death Many interviewees presented vivid and intensely felt memories of specific experiences with suicide bereavement funerals. This anxiety was exacerbated by an increased sense of wanting to ‘get it right’ for these families. 

We identified three themes capturing the distinguishing aspects of interviewees’ work with suicide bereavement funerals: Theme 1: Work & RoleTheme 2: EngagementTheme 3: Emotionality


**Theme 1: Work & Role: staff described their roles, the value of their work, the work environment and workflow and introduced the differences between funerals due to suicide.**


The company assigns the various tasks in the funeral process to staff in specific roles, from first contact by the bereaved to the funeral service itself. The workflow is standardised and centrally managed and administered.

Each role in the process is highly defined as and the following quotations illustrate interviewee descriptions of the characteristics of their work and roles in this context.


*Funeral Conductor:*
“Everyone’s got a section to bring that funeral together. For instance, the mortuary staff, they want to try and present that person … to make sure that is the best possible memory that you can have because some people get very damaged. So, there’s only so much you can do. The [Funeral Arranger] [puts] together the type of funeral that the family want at the location that they want, with everything involved in the funeral that they want, and then that gets handed on to the Conductor to make it happen on the day. The [Funeral Director’s Assistants] want to make sure that they’re there if they’re needed to assist. They want to make sure they’re on time. They want to make sure that they’re efficient. They want to make sure that they can do whatever around the family that needs to be done so it brings it all together. The Conductor, he [sic] wants to make sure the family are aware, or somebody around the family can guide the family [through the service], so that they know where they’re going. They know what’s going to happen next. You [as a Funeral Conductor] need to be able to flow it through.”

Interviews revealed that the significance and value staff placed on their roles and work was directly tied to addressing the wishes of clients. Staff explained the numerous forms of expertise and sensitivity this requires, all of which was largely learned ‘on the job’. In some cases, this was framed as ‘helping’ and ‘caring’ for their clients. In other cases, staff talked about the significance of providing guidance. For example, in the case of the funeral of a young person who has died by suicide, it will often be the first time their peers have attended a funeral service and they know little about the formalities. Interviewees advised that these funerals are typically large and highly emotionally charged, so staff may be required to support and guide young people on how to behave.


*Manager/Funeral Arranger:*
“The [staff at the funeral service] have a different sort of role particularly for a young person. [They are] dealing with obviously an incredibly grief-stricken family. The kids coming to the funeral are very emotional too [and] quite frankly they don’t quite know how to behave. There’s nothing in their skill set that’s given them any tools to know how to behave when they’ve lost someone close.”

Others spoke about how their professional expertise and knowledge can offer people security and comfort at a difficult time. Those involved in funeral preparations (Administration, Funeral Arrangers and Mortuary Staff) described the combination of sensitivity and experience they offer as especially important in the context of funerals due to suicide. 

A commonly used example was gently advising and re-directing clients away from requests that are unachievable or likely to create further distress such as when a family requests a private or public viewing of the deceased. The family (and mortuary staff) will want the presentation of the deceased person to be as close as possible to reality of the living person. However, if the death was due to injury and/or an internal autopsy has been conducted by the coroner there are likely to be visible scars or marks. In such cases, staff may advise the family to consider whether a viewing is appropriate, or what clothing should be provided to dress the deceased, to prevent further distress. 


*Funeral Conductor:*
“It’s more about the sensitivity of it … it’s about managing their expectations of what they’re going to see when they come and have a viewing. It is a conversation we have to have again and again. Sometimes it is about guidance and sometimes it is about just saying ‘I actually think this is not going to be in your best interests and this is why.’ Ultimately, it’s always going to be their decision. But you have to guide them, I think, at some stage and I’m comfortable with that. People come to us because we are the professionals and we know what we are doing.”


**Theme 2: Engagement: staff described adjustments in communication, practical tasks, timing and relational aspects of funeral planning and delivery.**


Interviewees were unanimous in saying that the very nature of a death by suicide, being sudden and often unexpected, fundamentally affected the manner in which the family respond to their loss and therefore, to the funeral process. It was the common experience of staff that most families bereaved by suicide have a lack of preparedness to engage in the funeral process. They said that this delineated these funerals from others in which there may also be sadness and complexity, but with an accompanying sense of the family’s acknowledgement of the ending of a life.


*Funeral Arranger:*
“The families are often traumatised and shell-shocked. So, it’s a very different feeling in the [Arrangement meeting] in comparison, say, to someone organising a funeral for their 90-year-old mother who’s had a long and fruitful life. Often, they can’t even articulate anything. There is just raw grief and emotion in the room and it’s hard to have a meeting like that. It’s a very long, drawn out couple of hours. It’s quite draining as well, from both sides, I think.”


*Funeral Conductor:*
“Look, a suicide has that extra layer because it was, for want of a better word, self-inflicted. Right? Whereas an accident they’re like—it’s still shocking. It’s sudden but it’s no fault of anyone. Do you know what I mean? It wasn’t purposely done. But the family is still in that situation where they’re—they’re in a situation they never thought they’d be in. They’ve been thrown into this situation and it’s the last place they want to be to me. So, it’s a very different feeling.”

Staff said that these funerals have a characteristic form of tension and distress. Additionally, that this required some adaptations on their part: not to the funeral process, which is the same for each client, but rather to their engagement with their clients—the family, next of kin and the deceased. These adjustments were subtle, and many staff described this experience as ‘walking on eggshells.’


*Funeral Conductor:*
“What I mean by the eggshells…[is that] you’ve just got to be careful. You really have got to have a look at the people in the family. You really have got to listen to what they do or don’t say. That can be a little bit eggshelly [sic]. That really can be. We might need to calm people down and we might need to take people out of the room because somebody just needs to vent, if you like. You get people come in when there’s been a suicide and they’re angry. And you have got to deal with that when it walks in that door. You’re the one that’s facing it.” 


*Funeral Director’s Assistant:*
“[At a suicide bereavement funeral] I get a sense of a real kind of ‘What do you say?’ ‘What do you do?’ You can read it on people’s faces, they want to talk about it. You hear all the whispers. As opposed to somebody who’s died of natural causes. ‘Oh, she had a great life.’ And it’s a great celebration of life for them. It’s a much happier mood at a funeral, if there’s such a thing. As opposed to—I think suicide is confusing, because people are still coming to terms.”

All staff reported that engagement and communication is less straightforward with families bereaved by suicide. They explained this as being due to a range of factors including shock and sensitivities around the matter of suicide itself. In their experience, families could be either blunt, open, or entirely silent about suicide being the cause of death. Together, these factors could create challenges in engagement with the family across the funeral process 


*Funeral Arranger:*
“In my experience families have just wanted to get it over and done with. They know they just have to do this arrangement. They just want it done. And they want to get out of there pretty quickly. They don’t want it to drag on. Because they’ve been thrown into the situation, they haven’t had time to think about what they would have liked to have done for a funeral. Whereas in other [expected] funerals I’ve had people wait two weeks because they want time to get it perfect. But with a suicide it seems everything is quite rushed.”

A key aspect of the tension experienced by staff was around ensuring they were using the right terminology with clients bereaved by suicide, in order not to cause further harm. Interviewees prioritised the knowledge and use of appropriate words and phrases to both inform and support their clients with the funeral process.


*Administrative Staff:*
“We’re more aware of language [when working with suicide] and I use the term ‘taking their own life’ [which] I think is a much nicer way of saying it. It doesn’t have quite as harsh an impact upon a person, I think.”


*Manager/Funeral Arranger:*
“It’s that delicate use of language and nuances and I’m always learning new ways of doing that. I will spend time pondering and working on the language…I like to have ways that I can discuss things without stumbling because I don’t think that’s kind and these people need us to be kind—honest but kind, not kind and unfaithful to the process.


**Theme 3: Emotionality: staff described the challenging intensity, unpredictability and complexity of emotionality present when working with suicide bereavement funerals and how they experienced and managed its impacts.**


Interviewees commonly reported that families bereaved by suicide have a more unpredictable expression of shock and grief than families in which a death has been expected. This could present as intense tears, anger, confusion, defensiveness, or detachment on the part of the family. Families had more or less openness to discussing the cause of death—and sometimes different members of families would take opposing views on whether to acknowledge the suicide. 


*Manager/Funeral Arranger:*
“There seems to be a degree of secrecy on the part of the family. Some of the families will state clearly what the cause of death was; suicide families might not so openly. Sometimes it comes up at some stage later when they relax. I never ask questions…But you can kind of sense it.”


*Funeral Arranger:*
“If someone passes away, they [a family bereaved by suicide] might of sort of say ‘he was found here.’ And you get this indication that you don’t want to go probing too much, because it seems like you’re doing it for just being nosy in a way. So yeah, people can be a little bit closed off and don’t really give much information.”

All interviewees said there was a greater emotional impact on themselves and their colleagues when working with suicide bereavement funerals compared with other funerals. This was concomitant with the baseline questionnaire in which nearly two thirds rated working with suicide bereavement funerals as having a greater emotional impact on other staff, and over half noted a greater emotional impact on themselves. 


*Manager/Funeral Arranger:*
“It [working on a suicide bereavement funeral] is more difficult for everyone emotionally.”


*Funeral Arranger:*
“The emotional intensity seems to be there quite strong, more so than with other funerals.”

All interviewees described at times sharing in the questioning, the shock, the anger, and the sorrow they observed in the families and the deceased with whom they worked. All interviewees reported cases in which they had shared a strong sense of their clients’ loss to suicide (particularly with funerals involving someone of an age similar to that of a loved one in their own lives) and experiencing some of the attributes commonly attributed to complex grief such as anger, confusion, and questioning. Many interviewees reported an increased sense of responsibility to, and protectiveness of, families and those deceased by suicide. This transference of feelings between staff and bereaved families, and staff and the deceased, are of great interest for further examination.


*Mortuary Staff:*
“I guess that’s one of our problems, as well. We get a little bit emotionally invested in a deceased person…especially with a young person, we just want them to look their absolute best. And we will beat ourselves up about and go above and beyond.”


*Funeral Arranger:*
“I was doing that. I was taking it away: I was feeling this could happen to me. What if this happens to my children. And you come away feeling quite low as well, hoping that it would never—that you’re never in that situation either.”


*Administrative Staff:*
“I think it’s just, you know, it’s a tragic end to a life. It was an unexpected end to a life. So, I think that’s why we kind of want to do our best for that person. Just a little bit more because you know that the family are really going to be grieving.”


*Manager/Funeral Arranger:*
“You sit with a family and you’re basically crushing their dreams because they had hopes and dreams for their loved one and you’re sitting there going, ‘So, we’re just going to look at a coffin now.’ And that’s just—you’re just a bearer of bad news and you talk about grave debts [cemetery fees] and they just look at you like this is the last thing they want to be doing today and sometimes you are the worst person in the world because you are forcing them to consider things that they never in their wildest dreams ever thought they would have to consider.”

Staff described the emotional labour [32] of their work in the context of suicide bereavement funerals as; ‘walking on eggshells’ (also raised in the Engagement theme), ‘holding back a bit’, ‘not wanting to intrude’, ‘draining’, using ‘limits’ or ‘levels’ as well as wanting to be well prepared. They said these adjustments were made individually, without formal consultation or direction. 


*Funeral Arranger:*
“There’s a level…that’s where you’ve got to make sure that you’ve got that, say, level you can go so far, engage yourself to a certain level but then you’ve also got to make sure that you’re taking a step back a little bit to make sure—because otherwise you can take everything home with you and it can play on your own mind as well.”


*Funeral Conductor:*
“You hear it’s a suicide and then—so yes, you do think, ‘Well, hang on a minute’. Okay, so you do pull back a little bit. You want a little bit—you want to try and get as much information as you can, but [also to] just to maybe prepare yourself for the family. Because you don’t know what you’re going to get. And you know why that is? It’s probably—and I mean I don’t even know if this is true because maybe it’s just the way that—after doing so many funerals. It’s the fact that it wasn’t an illness, and it wasn’t expected, and no one knew this was going to happen. It was suicide. It is different. It just is. It just is. So we need to just remember that. We need to prepare ourselves.”

There was acknowledgement that this emotional labour was not without its long term risks to staff wellbeing. Interviews emphasised that if they were to ask for help with this, staff preferenced informal peer support over formal counselling services. 


*Manager/Funeral Arranger:*
“I think these jobs and this profession is prone to burnout. The people who are not looking after themselves by stepping up and switching off they really burn out and they either—they usually quit the job or step down from the role or something.”


*Mortuary Staff:*
“My job is dangerous, not from a physical aspect, but from an emotional and mental aspect as well. If you don’t deal with it, you don’t—if you can’t process those—if you hold on to things like that, it can damage you. You know? And that’s something that—you see people who’ve worked in the industry for a couple of years, you see little cracks appearing in personality, how they deal stuff, attitudes.”

### 3.5. Responses to the Training

Almost all the interviewees referred to the first benefit of attending the Training as being able to share their experiences and learn from colleagues working in different roles across the organisation. They also appreciated the information and resources relating to appropriate language and communication and resources relating to suicide bereavement. A small number of interviewees indicated that the Training was useful in terms of their own thinking and approach to suicide bereavement funerals.


*Funeral Arranger:*
“My journey with suicide has probably matched with a lot of people in the community. It took me years and just experience, and [the] Training got my head around to the space that that’s their right, it is their own life and sometimes it’s the only alternative they can consider. So, who am I to judge that?” 

Staff also reported improved openness and willingness to acknowledge the challenges of these funerals by the Company. In a broader organisational context, interviews showed that the Training had helped open a conversation among staff and at management level about the common challenges presented by the work with suicide bereavement funerals. 


*Funeral Arranger:*
“I think [post-Training] there is far more openness [about working with suicide bereavement funerals] which is very healthy for everyone.


*Manager:*
“What we didn’t do well for years is to highlight to staff that they are going to a funeral for somebody who has taken their own life. I was told when I started that it was none of our business and that had all sorts of issues tied into it because suicide was still a taboo, and some families didn’t want you to know. I do it [inquire about if the loss might be due to suicide] much more comfortably now because of the [Working Well with Suicide Bereavement Funerals] Training.” 

As a result of the Training and engagement with the research process, company management had considered processes to mitigate the identified challenges of these funerals through workflow adjustments (finding ways to reduce repeated assignment of staff to funeral due to suicide in short periods), communications (finding ways to improve staff preparedness for these funerals) and practices (acknowledgement of staff taking requiring leave due to their own experiences of suicide bereavement) including considering practices related to Work Health and Safety.

### 3.6. A Model of Workplace Postvention

As noted in the Introduction, despite the extent of the impact of suicide bereavement in Australia (Maple et al., Op Cit) the evidence base for postvention models is limited [33]. This Case Study demonstrated that workplace postvention approaches could be usefully informed by engaging with the ecology of individual workplaces, understanding their particular contexts, staff experiences and requirements. This method reflects the characteristics of a public health approach to postvention as outlined by Andriessen et al. [34] and which this study suggested might be applied as follows:Understanding workplace and staff needs in the context in which they operate;Applying postvention research and practice to address identified needs;Delivering bespoke postvention components (training, governance, resources and integration/networks) to support preparedness and responsivity; andImplementation of ongoing research and evaluation for organisational and program improvement.

The proposed approach is in clear distinction to current postvention guidelines which require users to adapt to meet generic criteria without appropriate support, ongoing review, or integration with other stakeholders in the environment in which they operate. Rather, this approach would utilise lived experience of staff together with current postvention research and evidence to develop and integrate targeted interventions informed by, and appropriate to, the system in which they were applied. This approach is illustrated in Figure 1.

This Case Study indicated that this approach to workplace postvention is both generative (providing details on staff experiences and needs to which the Training could respond) and developmental. That is, over the period in which the organisation engaged with the Training and research process, management identified the potential for other postvention components. For example, following the conclusion of the Case Study, the organisation has requested support with postvention governance including operationalising legislated work health and safety (WHS) related requirements for managing psychosocial hazards in the workplace. 

In terms of a workplace postvention model, the Case Study findings were clear that beyond staff training, organisational governance (organisational policy and processes, e.g., staff induction, supervision, mentoring, WHS, etc) was required to support good practice and ongoing improvement. In addition, the research suggested that the strength and sustainability of the training and governance components could be extended through broader integration with relevant external supports and stakeholders. In the context of this Company, that may include networking relationships with other funeral providers and related agencies including the State Coroner, State Mortuary, first responders, suicide prevention and postvention services and suicide bereavement counselling services. This model is illustrated in Figure 2.

## 4. Discussion

This mixed methods Case Study included baseline and post-training questionnaires and interviews with a study sample of 67 funeral staff working in a range of roles in a family-owned metropolitan funeral company. 

The study established that funerals due to suicide present a greater level of difficulty for staff together with a greater level of emotional impact. This is not because suicide bereavement funerals are organised differently. In practice, each of the stages in the workflow, role and task requirements are the same as with any other funeral. However, staff experience suicide bereavement funerals as more emotionally demanding and complex in terms of communication and engagement with the bereaved and mourners.

Results showed that the question of ‘why’, so closely attached to loss by suicide [35], was also taken up by staff as they worked with these funerals. They asked it of themselves and reflected on what might happen if it occurred in their own families. They asked it of the deceased in terms of what might have caused their death or protected them from suicide. They universally expressed great empathy for the shock, intensity, and range of emotional reactions of the bereaved. The ‘eggshells’ staff described across the ‘Engagement’ and ‘Emotionality’ themes were not just about negotiating the increased emotional intensity of those bereaved in this manner. The ‘eggshells’ also represented their own heightened concerns about the family, their sense of responsibility for how and why the bereaved were responding, their questions about how and why the deceased had taken their life, how the sometimes injured body (self-injured and/or injured through internal autopsy) would present at a viewing, and the way the family responded throughout the funeral process. The responses of staff reflect those of the families they described and are also common to the experience of suicide bereavement. In a systematic review of grief reactions comparing bereavement by suicide with other causes of death, those with a loss to suicide reported higher levels of rejection, shame, stigma, need for concealing the cause of death and [self] blaming than all other groups [36]. This complex response has also been cited in research on the impact of the death of a colleague by suicide [37] and amongst psychiatrists and psychologists following the death of a patient by suicide [38]. 

These findings suggest that there is a unique form of ‘emotional labour’ integral to funeral work and more specifically, with funerals due to suicide. In a broader sociological context, the matter of suicide is highly contentious. It attracts social, religious and cultural stigma that extends to the person experiencing suicidal thoughts, to those who have died by suicide and to their families [39]. In Goffman’s terms [40], they are set apart in a manner they cannot control and to which they cannot respond, and thus, they turn inward. Suicide bereavement can be characterised as a form of (socially) ‘Disenfranchised Grief’; that is, “grief that persons experience when they incur a loss which is not or cannot be openly acknowledged, publicly mourned or socially supported.” [41] (p. 1).

The notions of stigma and disenfranchised grief are helpful in understanding the context of funeral work and for considering how staff manage their responses to suicide bereavement funerals. An ethnographic study of funeral staff in the United States showed that “funeral directors are acutely aware of the stigma associated with their work, most of which comes from handling the dead and being viewed as profiting from death and grief” [42] (p. 403). Thompson posits that funeral staff attempt to neutralise this perceived stigma by “practicing role distance, emphasising professionalism, cloaking themselves in the ‘shroud of service’ [and] providing important and making necessary services for the living” (Op Cit p. 403). This Case Study shows funerals due to suicide present a direct challenge to this arrangement. Just as families bereaved by suicide experience their loss differently, staff experience and are differently impacted by this type of funeral. 

The staff in this Case Study had learned to respond to the challenges of suicide bereavement funerals without formalised training or support. The ‘limits’ staff described putting in place for themselves were not taught, trained, or even openly discussed in the Company; they were learned and developed on the job. They were described by staff as fundamental to their work and roles.

This lack of formalised support contrasts with caregiving professions, which have established processes and structures to support staff working with human distress. For example, registration as a psychologist in Australia requires a minimum of six years’ education, two years practicum, and ongoing training and supervision [43]. 

At an organisational level, the ‘eggshells’ around suicide bereavement funerals were embedded within the historical practice of refraining from asking about, or discussing, the cause of death. This effectively required that staff find their own ways to manage the challenges and impact of this work. The organisation design, which differentiated staff contact (with clients and with other staff) according to role and function, further reinforced this requirement as did the broader sociological context of attitudes to death, funeral work and suicide. 

This study demonstrated that the benefits of staff informed postvention training could be extended and embedded through the introduction of a workplace postvention model incorporating governance and integration. These findings are of relevance to the broader funeral industry and could be further enhanced by investigating the potential of the proposed workplace postvention model and its broader application to other industries and workplaces.

### Limitations

Regarding the sampling method, there are some caveats to consider in terms of the broader population of funeral service providers: the data were obtained from 67 staff working in a family-owned funeral company; it was based in a metropolitan area rather than a regional community; it has a forward-thinking leadership willing to engage with this research and training (the researcher is unaware of similar training delivered in Australia or internationally); and like any workplace, it has its own personality or culture that is unlikely to be generalised to all funeral service providers.

## 5. Conclusions

This study explored the experience of funeral staff working with funerals due to suicide. As the first formal service providers contacted by Australian families and friends following bereavement, findings revealed the challenges these staff face and how these may be mediated though the introduction of staff informed postvention-training.

Through their improved awareness and confidence, these staff trainees may take a positive part in a public health approach to suicide postvention. If extended to other funeral providers, the Training could contribute to a reduction in stigma around suicide bereavement, increased community awareness of appropriate bereavement services and supports, and potentially a decrease in the risk of further harm among those impacted by suicide.

However, in a public health context, training any staff group to be better informed and aware is only one step towards postvention. This organisation is only one of many in which staff are responding to suicide in the absence of the targeted coordination and support which could be offered through the introduction of organisational postvention practices. This exploratory study could be further enhanced by investigating the non-training components of the proposed workplace postvention model to understand their application and efficacy.

## Figures and Tables

**Figure 1 ijerph-20-00142-f001:**
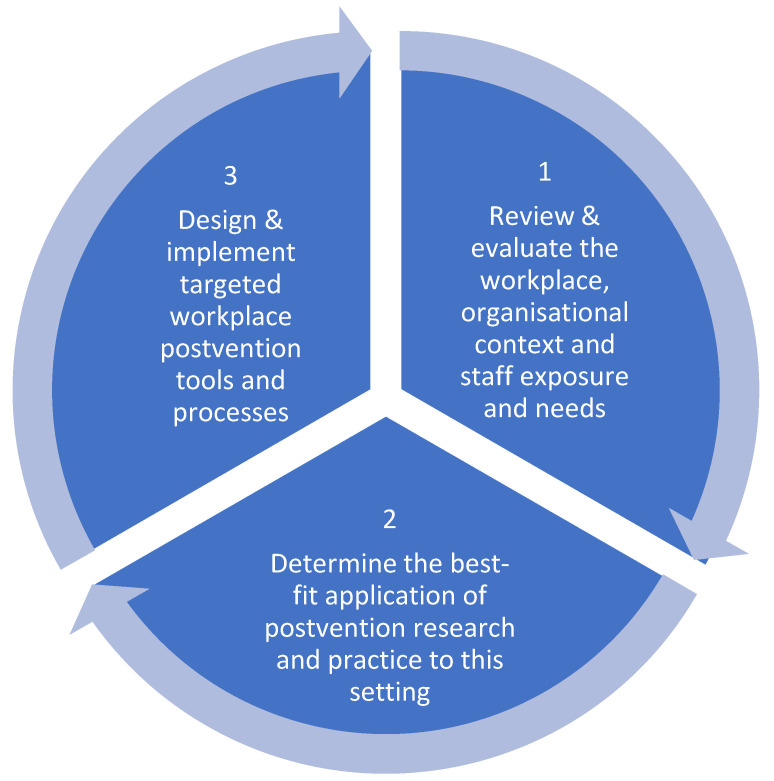
A Public Health Approach to Workplace Postvention (2022).

**Figure 2 ijerph-20-00142-f002:**
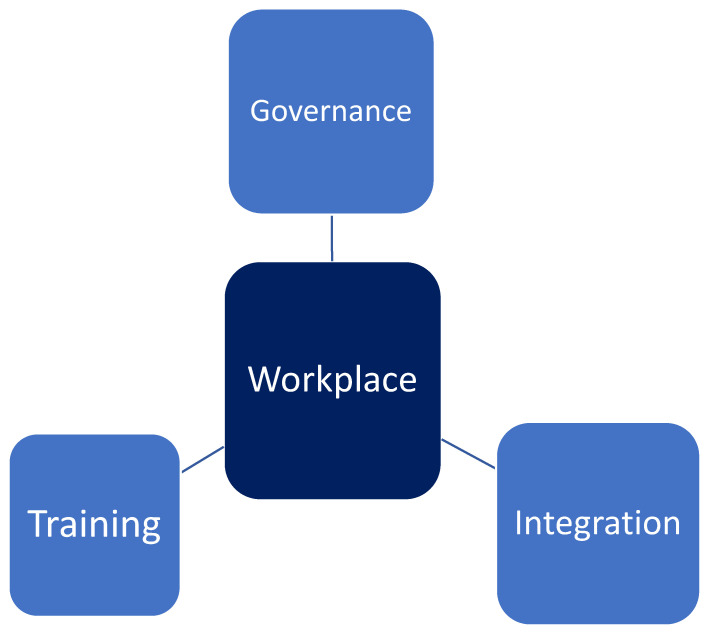
A model of workplace postvention (2022).

**Table 1 ijerph-20-00142-t001:** Baseline questionnaire results (*n* = 67).

Variable	Less than Other Funerals No. (%)	Same as Other Funerals No. (%)	Greater than Other Funerals No. (%)	Not Worked on Suicide Bereavement Funeral No. (%)
Level of difficulty	0 (0)	14 (20.9)	50 (74.6)	3 (4.5)
Emotional impact on other staff	0 (0)	25 (37.3)	42 (62.7)	-
Emotional impact on self	0 (0)	26 (38.8)	38 (56.7)	3 (4.5)
Difficulty finding the right words to use	0 (0)	11 (16.4)	52 (77.6)	4 (6)
Overall confidence	11 (16.4)	49 (73.1)	7 (10.5)	-
Overall comfort	18 (27)	43 (64.2)	6 (9)	-

**Table 2 ijerph-20-00142-t002:** Post-Training Questionnaire results (*n* = 32).

Variable	No Change No.%	Somewhat Useful No.%	Very Useful No.%
The training and resources have been useful in working with suicide bereavement funerals *	-	14 (43.8)	17 (53.1) *
The training and resources have helped you better understand the emotional impact of working with suicide bereavement funerals	-	16 (50)	15 (46.9)
The training and resources helped you better manage the emotional impact of working with suicide bereavement funerals	3 (9.4)	16 (50)	12 (38)
The training and resources helped reduce the amount of pressure you feel working with suicide bereavement funerals	14 (43.8)	15 (46.9)	2 (6.3)
Following the training, please rate your level of confidence that you know/use appropriate language working with suicide bereavement funerals	4 (13)	15 (46.9)	12 (38)
Please rate your awareness, following the training of suicide bereavement and support services	3 (9.4)	13 (40.6)	14 (44)
Please rate your level of comfort, following the training, in sharing resources about suicide bereavement support services with families and mourners	1 (3.1)	21 (65.6)	9 (28.1)

* One staff member had not worked on a suicide bereavement funeral and a number of questions were unanswered.

## Data Availability

This study was completed as part of an MPhil Dissertation in the School of Allied Health at the University of WA and is available from aliclements.consulting@gmail.com.

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
