# Peer review of "Towards an Evidence-Based Model of Workplace Postvention"

_ijerph, 2022, doi:10.3390/ijerph20010142_

Round 1
Reviewer 1 Report
Dear authors, I have attached a document detailing my comments and suggestions. I believe your study to be novel and important in the field of postvention research, and I have aimed my feedback to support your work in developing a more robust, well evidenced and well-presented argument.

Author Response
Thank you for your very thoughtful and helpful review. I have addressed your suggestions and outlined all changes in the attached document.

Reviewer 2 Report
In research methods, quantitative analysis seems to be lacking. However, I think it is an interesting enough study for the reader.
It is considered that there are contents that can be provided not only by funeral companies but also by local governments and central governments. It is explained through Figures 1 and 2, but it would be a better study if the authors suggested roles that funeral companies, local governments, and central governments can play in addition to these contents.
Thanks to the authors for submitting an interesting study.
Author Response
Thank you for your review. I have referenced and discussed government frameworks and approaches to suicide prevention and postvention in the article and also how health and other industries deal with postvention. I trust this will address your helpful suggestions.
Round 2
Reviewer 1 Report
Dear Authors. Thank you for your clear and comprehensive response to my previous review of your manuscript. I would like to commend you on the efforts that you have made and which I believe greatly improve and strengthen your work. In particular, the re-positioning of the two models seems to give them greater visibility and importance in the manuscript as they now lead on from your findings. I believe that Figure 1 is an excellent contribution to the field of postvention in providing a very useful and important framework for the development of guidance. Additionally, your discussion section has really come to life, the re-working and inclusion of numerous citations adds strength to this work, and to the case that you are putting forward. I am recommending that this manuscript be accepted for publication pending a few minor typos which I list here.
Line 72: remove 'perhaps'
Line 72: change 'and offering' to 'which offers'
Line 78: change evidenced to evidence
Line 171: remove 'via'
Line 189: change 'representing' to 'represent'
Line 240: remove 'a'
Line 377: remove the first 'staff'
Line 407: Remove extra space between sentences
Line 582: insert 'x' into 'eternal' to read 'external'
Line 577: I suggest starting a new paragraph at 'In terms of...'
Line 657: remove 'a'
Finally, I suggest putting the limitations section before your conclusion. Congratulations on this manuscript, it makes a useful and important contribution to the field.
Author Response
Thank you again for your time, attention, patience and encouragement. I feel so very lucky to have had you review my first paper! Each of the last edits have be made as recommended.
Warm regards
Ali Clements